# Stress/Immune Biomarkers in Saliva among Children with ADHD Status

**DOI:** 10.3390/ijerph18020769

**Published:** 2021-01-18

**Authors:** Anna Krahel, Elzbieta Paszynska, Agnieszka Slopien, Maria Gawriolek, Justyna Otulakowska-Skrzynska, Szymon Rzatowski, Amadeusz Hernik, Tomasz Hanć, Ewa Bryl, Paula Szczesniewska, Karolina Bilska, Joanna Duda, Marta Tyszkiewicz-Nwafor, Monika Dmitrzak-Weglarz

**Affiliations:** 1Department of Integrated Dentistry, Poznan University of Medical Sciences, 60-812 Poznan, Poland; akrahel@wp.pl (A.K.); mgawriolek@gmail.com (M.G.); justynao@ump.edu.pl (J.O.-S.); szymonrzat@gmail.com (S.R.); amadeusz.hernik@gmail.com (A.H.); 2Department of Child and Adolescent Psychiatry, Poznan University of Medical Sciences, 60-572 Poznan, Poland; asrs@wp.pl (A.S.); malamt@gmail.com (M.T.-N.); 3Institute of Human Biology and Evolution, Faculty of Biology, Adam Mickiewicz University, 61-614 Poznan, Poland; tomekhanc@gmail.com (T.H.); ewabryl2@gmail.com (E.B.); mamrotpaula@gmail.com (P.S.); 4Psychiatric Genetics Unit, Department of Psychiatry, Poznan University of Medical Sciences, 60-806 Poznan, Poland; kbilska@ump.edu.pl (K.B.); joanna.duda91@gmail.com (J.D.); mweglarz@ump.edu.pl (M.D.-W.)

**Keywords:** ADHD, saliva, cortisol, alpha-amylase, IgA, IgG, IgM, overweight

## Abstract

*Background*. This cross-sectional study aimed to evaluate stress and immune biomarkers in saliva samples of attention-deficit/hyperactivity disorder ADHD compared to healthy non-ADHD children. *Material and methods*. A total of 132 children under 11 years old (8.5 ± 1.1) enrolled in a cross-sectional study: with confirmed ADHD (n = 60) and healthy controls (n = 72). The clinical evaluation included physical measurements (height, waist, hip circumference, body weight, body mass index BMI, BMI z-score) and unstimulated saliva collection and measurements of free cortisol, salivary alpha-amylase (sAA), and secreted immunoglobulins (sIgA, IgG, and IgM) with quantitative assay (ELISA) analysis. Unpaired *t*-test, Welch test, or Mann–Whitney U test were applied for group comparisons when appropriate, and the correlation between variables was analyzed with Spearman’s rank coefficient. Results were considered significant at *p* < 0.05. *Results*. In the ADHD group, body weight (*p* ≤ 0.01), BMI (*p* ≤ 0.009), and hip circumference (*p* ≤ 0.001) significantly differed, while waist size and BMI z-score did not (*p* > 0.05). Significant elevation of the salivary sAA (*p* = 0.03), sIgA (*p* = 0.02), and IgM (*p* ≤ 0.001) biomarkers were detected, without differences in the morning cortisol (*p* > 0.05). Significant correlations between cortisol and BMI, hip size, and IgA, as well as between IgG and sAA and IgA were obtained. *Conclusions.* Saliva can be used to monitor ADHD status with regard to biomarkers indicating the hypothalamus–pituitary–adrenal axis, as HPA axis, and sympathetic activity. The results indicate that morning collection of saliva in contrast to unchanged salivary cortisol, may evaluate mentioned above system dysregulations by measurements of sAA and immunoglobulins among ADHD children.

## 1. Introduction

A dental office is a frequently visited health-care facility by children and adolescents under 18 years of age. This allows easy contact with children and may also help in monitoring mental stress during the developmental period of life among healthy or unhealthy subjects. 

Diagnosing and managing attention-deficit/hyperactivity disorder (ADHD) in childhood are exceptional achievements for medical clinicians [1]. Research has shown that, for children with ADHD, there is not an established, adequate diagnostic related to clinical symptoms, and they may suffer from significant dysfunction [2,3]. Some of the ADHD symptoms, such as extreme hyperactivity, may appear or disappear over time. Additionally, obtaining information from the family could be limited [1]. The abovementioned difficulties are only examples of scientific gaps in research that may affect the ability to screen, diagnose, and treat ADHD. A critical level of stress and inflammatory biomarkers in saliva among young ADHD individuals could be a source of potential knowledge to determine the diagnosis and course of ADHD disease empirically [4]. 

The salivary glands work together with the neuroendocrine system, which makes saliva sensitive to stress or mood behaviors [5]. Valuable stress indicators such as salivary cortisol, salivary alpha-amylase (sAA), secretory immunoglobulin A (sIgA), G (IgG), and M (IgM) can commonly be found in saliva as well as in other body fluids. Still, as a response to mental stress, the largest variability in these biomarkers’ concentrations, may be recorded in saliva [6,7]. 

Many studies have described a significant effect of stress on the activation of the hypothalamic–pituitary–adrenal (HPA) axis, resulting in increased cortisol production [6,7]. Cortisol is released according to a person’s circadian rhythm, with a peak in the morning after waking up. Its level decreases during the day to reach the lowest value just before bedtime [8]. Exposure to a stress factor is associated with an increase in cortisol production and a rapid return to its resting level [8]. Chronically sustained elevated or lowered cortisol levels may lead to mood disorders [9,10], social isolation [8], eating disorders [11], and neurodevelopmental disorders [12]. 

Currently, it is proposed that the HPA axis and the activity of the sympathetic nervous system (SNS) have to be monitored in parallel during mental stress [13,14,15]. Saliva can be a source of information on SNS activity through salivary alpha-amylase concentration [16]. Complementary, stress-induced alterations in saliva were detected in other groups of patients with mental disturbances [17,18]. 

Just as salivary cortisol and amylase have been studied quite frequently, the salivary immunoglobulins of the immune system, such as IgA, IgG, and IgM, have been unexplored in ADHD [19]. The oral mucosal immune system is one of the defense effectors against human pathogens and also reveals a subclinical immune reaction to hyperactive behavior in ADHD [20]. The measurement of salivary immunoglobulins might be indicated as a non-invasive method to determine the immune status of ADHD children in terms of general and oral infections. Investigations focusing on stress/immune biomarkers and ADHD are few. An examination of the abovementioned biomarkers may at the same time provide a comprehensive knowledge about all aspects of stress regulation [21,22,23,24]. Therefore, this study aimed to evaluate salivary cortisol, sAA, IgA, IgG, and IgM among children with clinically confirmed ADHD against healthy, non-ADHD children.

## 2. Material and Methods

### 2.1. Sample of the ADHD Group

The intention was to select children with ADHD from one part of the area, the western sector of the country (Appendix A). The recruitment of children with suspected ADHD was announced in GP clinics (n = 100) and mental health clinics for children and adolescents (n = 20). In total, 69 families having a child with a suspected ADHD diagnosis applied for recruitment and were willing to include their child in the investigation. Ultimately, in the 69 children, a diagnosis of ADHD was determined by two specialists, a psychiatrist and a clinical psychologist, according to the diagnostic criteria of DSM-V [25,26]. Most of the examined children had symptoms from all criteria areas of the disorder, which allowed for the diagnosis of a mixed form of ADHD. Among the abovementioned children who came for the study, the diagnosis was confirmed in 64 of them. Two children were excluded due to a lack of cooperation during a dental examination in the office, and the parents of two children withdrew their consent. Ultimately, the study group consisted of 60 children aged 6 to 11 years (8.6 ± 1.2) with a confirmed diagnosis of ADHD: 50 boys (82.1% of all participants in the study group) and 10 girls (17.9%) (Appendix A).

#### Sample of the Control Group

In one agglomeration, four primary schools were randomly selected from opposite geographic regions––eastern, western, northern, and southern. In the schools, the project was proposed to parents of children aged 6 to 11. The researchers offered a free child-development analysis and dental examination at the dental office. Four hundred willing parents were enrolled from four schools to have their children examined. Regarding the inclusion criteria (Table 1), 150 children were pre-qualified for further evaluation. After the general interview, 50 children did not meet inclusion age criterion because they were over 12 years old. In addition, 18 children did not cooperate during the dental examination or did not attend school for more than 4 weeks. Finally, 72 children were allocated to the control group, with ages ranging from 6 to 11 years (8.3 ± 1.1); 45 boys (62.5% of the control group) and 27 girls (37.5%) were recruited among primary schoolchildren (Appendix A).

### 2.2. Ethical Approval

The cross-sectional project involved non-invasive analysis methods such as salivary sampling and questionnaires to confirm or deny a diagnosis of ADHD. The study plan was correct according to the standards of Good Clinical Practice (GCP) and had a positive opinion of the local Ethics Committee of the Medical University of Poznań (Resolution No. 26/18) [27]. All parents and children were informed about the aims and methods of the research, and their legal representatives gave their written consent.

### 2.3. Assessment of Child Development

The anthropometric assessment of a child’s development consisted of body measurements, such as height, waist, hip circumference, and body weight. The standing body height was measured with a SECA 216 wall-mounted stadiometer with an accuracy of 0.1 cm. The body weight of children was assessed in light clothing on a digital scale with an accuracy of 0.1 kg. The waist circumference was determined at the midpoint between the lower edge of the costal arch and the upper iliac crest using a metric tape [28]. The hip circumference was measured parallel to the floor and taken as the greatest circumference of the buttocks using a metric tape [28]. All subjects were assessed according to Body Mass Index (BMI), age, weight, and height. The BMI was transformed into z- scores based on WHO growth charts [29]. 

### 2.4. ELISA Salivary Samples

All participants in both groups underwent the same procedures in the dental office to avoid any bias that could affect salivary flow during the examination. The patients were examined in the same season of the calendar year (fall/winter) and at the same time in the morning, i.e., between 9:00 and 10:00. The same examiner took the saliva samples. All patients were examined 60 min after breakfast and oral hygiene procedures. Unstimulated total saliva was spat into sterile containers for 15 min. Children were instructed to move their body to forward position and focus on the spitting activity. The first portion of the saliva was collected in a separate container and then allowed to accumulate. It was then spit out into a sterile disposable tube [30]. Immediately after the collection, the container was sealed and subjected to centrifugation, supernatant separation, and two-stage freezing at −20 and then −80 °C. The tests for salivary biomarkers were performed as follows.

#### 2.4.1. Salivary Cortisol

The concentration of free cortisol in the saliva was quantitatively determined by the ELISA method using the ELISA Kit DES6611 for the In Vitro Diagnostic (IVD) (Demeditec Diagnostics GmbH, Kiel, Germany). The standard curve ranged from 0 to 30 ng/mL, the intra- and inter-assay variability coefficient was assessed to be below 6%, and the standard curve was statistically significant (r^2^ = 0.998, *p* < 0.001) [31,32]. 

#### 2.4.2. Salivary Alpha-Amylase (sAA) 

The sAA concentrations were quantitatively determined by enzyme-linked immunosorbent assay (ELISA) using the ELISA Kit DEEQ6231 for the In Vitro Diagnostic (IVD) (Demeditec Diagnostics GmbH, Kiel, Germany). The standard curve ranged from 0 to 500 U/mL, the intra- and inter-assay variability coefficients were assessed to be below 5%, and the standard curve was statistically significant (r^2^ = 0.982, *p* < 0.001) [33].

#### 2.4.3. Salivary Secretory IgA (sIgA)

Salivary sIgA concentrations were quantitatively determined by the ELISA method using the ELISA Kit DEXK276 for the In Vitro Diagnostic (IVD) (Demeditec Diagnostics GmbH, Kiel, Germany). The standard curve ranged from 0 to 400 µg/mL, the intra- and inter-assay variability coefficient were assessed to be below 5%, and the standard curve was statistically significant (r^2^ = 0.995, *p* < 0.001) [34,35].

#### 2.4.4. Salivary IgG

Salivary IgG concentrations were quantitatively determined by the ELISA method using the ELISA Kit E0544h for research use only (RUO) (EIAab, Wuhan, China). The standard curve ranged from 0 to 5000 pg/mL, the intra-assay variability coefficient was <4.4%, and the inter-assay variability was <7.8%, respectively (r^2^ = 0.999, *p* < 0.001) [36,37]. 

#### 2.4.5. Salivary IgM

Salivary IgM concentrations were quantitatively determined by the ELISA method using the ELISA Kit E0543h for research use only (RUO) (EIAab, Wuhan, China). The standard curve ranged from 0 to 500 ng/mL, the intra-assay variability coefficient was <6.2%, and the inter-assay variability was <9.1%, respectively (r^2^ = 0.986, *p* < 0.001) [38]. 

All ELISA tests were performed according to the manufacturer’s instructions without any modifications. All samples and standards were run in duplicate, and the mean value of the two assays was used for statistical evaluation. Optical density was read with a spectrophotometric plate reader (Asys UVM 340 Microplate Reader from Biochrom Ltd., Cambridge, UK) for a wavelength of 450 nm ± 10 nm. A four-parameter algorithm (four-parameter logistic) was used to assay the concentration in the tested samples.

### 2.5. Statistical Data Analysis

The analyzed data were expressed as mean ± standard deviation, median, and minimum and maximum values as appropriate. The normality of the distribution was tested using the Shapiro–Wilk test, and the equality of variances was checked using Levene’s test. A comparison of two unpaired groups was performed using the Mann–Whitney U-test. The relationship between variables was analyzed with Spearman’s rank correlation coefficient. All results were considered significant at *p* < 0.05. Statistical analyses were performed with STATISTICA v13 software (StatSoft Inc., Tulsa, U.S.).

## 3. Results

### 3.1. Sample

The final sample consisted of 132 participants: (60 ADHD and 72 control children) (Table 1). The mean age of the ADHD children was 8.6 ± 1.1 and that of the controls was 8.3 ± 1.2 years with no statistically significant difference for age and height (*p* > 0.05). The ADHD participants weighed 10.5% more than those in the control group (*p* ≤ 0.01), as body weight (*p* ≤ 0.01), BMI (*p* ≤ 0.009), and hip circumference (*p* ≤ 0.0002) were statistically different between patients and controls, but not the BMI z-score or waist size (*p* > 0.05) (Table 2). 

#### 3.1.1. Stress and Immune Biomarker Levels

An analysis of the salivary samples revealed a significant elevation of the sAA (*p* = 0.03), sIgA (*p* = 0.02), and IgM (*p* = 0.0000001) levels in the ADHD group (Table 2). However, no differences in the free-cortisol level were detected between the ADHD and healthy children of the control group (*p* = 0.739) (Table 2). 

#### 3.1.2. Correlations

In the ADHD group, the Spearman analysis showed a significant correlation between cortisol and BMI (r_s_ = −0.35; *p* = 0.01), hip size (r_s_ = −0.28; *p* = 0.04), and IgA (r_s_ = 0.35; *p* = 0.01). IgG was correlated to alpha-amylase (r_s_ = −0.38; *p* = 0.005) and IgA (r_s_ = −0.47; *p* = 0.0005) (Table 3). 

In the control group, significant correlations were found only between IgA and IgM (r_s_ = −0.308; *p* = 0.01) (Table 3). 

## 4. Discussion

The selected salivary biomarkers were chosen by evaluating previous studies on salivary stress and immune biomarkers and ADHD. However, there are limited results in the scientific literature, mostly connected with cortisol and sAA but not IgA, IgG, or IgM [19]. Saliva proved to be a diagnostic fluid to obtain information about the state of the children’s health, but not without significance is the non-invasive nature of the examination [39]. Our results showed similar cortisol concentrations in the groups of children with and without ADHD. Previously published investigations, based on 24-h cortisol monitoring in children with ADHD suggested a different regulation of the hypothalamic–pituitary–adrenal axis with a decrease concentration, especially in the morning [12,40]. In subsequent studies, children with ADHD revealed lower cortisol levels in the morning and at bedtime than during the day [41,42]. Therefore, coordination and control of cortisol levels with abnormal fluctuations throughout the day were considered. Unbalanced peaks and decreases in cortisol secretion and its irregular continuity towards increased concentrations may have affected the reversal results [40,41,42,43,44,45,46,47]. The hypothesis that bad life experiences, permanent stress, and adversities in early childhood may affect the HPA axis, leading to a decrease in cortisol levels in ADHD, has not been confirmed, except for a positive association between childhood adversity and a morning cortisol rise [42]. Therefore, the lack of significance in the cortisol level between the groups is justified because cortisol is characterized by a relatively high variability in secretion after the stress factor applied within a few minutes. It then quickly changes its level and returns to the daily profile. Even if children show increased cortisol concentration during the day, its liberation is accelerated at night [48,49]. The abovementioned mechanisms of cortisol variability and further analyses of its level in children with ADHD would require assessments over a 24-h period.

Nevertheless, determinations in saliva open many paths due to the non-invasiveness and repeatability of the measurements [50,51].

The results in this study of an increased sAA concentration compared to cortisol found raised some questions and notes. In the analyses of other authors, the sAA was collected during acute and chronic stress stimuli, and higher concentrations were obtained [52,53].

In one ADHD study, a lower evening and diurnal sAA secretions in response to stress were noted [54]. It should be remembered that the salivary glands are innervated by both the sympathetic and parasympathetic parts of the autonomic nervous system (ANS). Therefore, its innervation suggests changes in response to stress [7]. It should also be underlined that children with neurodevelopmental disorders may show atypical stress responses and have alterations in their diurnal profile of stress biomarkers. In our study, the significantly demonstrated level of sAA suggests that hyperactivity of the HPA axis in subjects with ADHD may be accompanied by disturbances in the regulation of the ANS [44]. 

In the present study, all immune biomarkers, IgA, IgG, and IgM, were significantly increased in the salivary samples of ADHD children. The level of secretory IgA was markedly higher among ADHD children than in those in the control group and correlated to free cortisol and IgG levels in the saliva. This finding shows that sIgA levels may be linked to the stress response and may also confirm a link to HPA axis hyperactivity. According to the current literature, an increase in IgA in oral saliva may also indicate a greater hypersensitivity of the oral mucosa and susceptibility to infectious diseases [55]. This increase in IgA levels can be attributed to high oral responsiveness to injuries, gingivitis, and an increased concentration of yeast and *Lactobacillus* [56]. Additionally, overall our results may help to interpret other research findings relating to ADHD and immune responses between children and adolescents [20]. It could also indicate pathological oral conditions, especially those that are infection related [57]. 

Our research suggests many directions in which salivary enzymes and immunoglobulins may be analyzed. Their concentrations in saliva are essential also for stimulating dental caries development in the oral cavity. Our previous study showed high dental caries in primary and permanent dentition among ADHD children [58]. Additionally, tooth decay was related to sweetened drink intake [58]. Therefore, alpha-amylase and immune biomarkers may both be at an enhanced level, and further research is necessary to link oral homeostasis and ADHD.

Our results showed a significant increase in body mass, hip circumference, and BMI with correlation to cortisol. Comparative research also indicates excessive body mass or BMI compared to pediatric norms or control groups [59,60,61,62]. The ADHD–obesity relationship can be explained by genes [63], neurobiological features [64], deficits in executive functions [65], fetal programming [62], sleep disorders (circadian rhythm), incorrect eating habits [66], high-fat diet [67], reduced physical activity [61,68], and stress [69], as well as a sedentary lifestyle. It can be assumed that poor stress resistance affects physical health in overweight children with ADHD [70]. 

## 5. Conclusions

This cross-sectional study aimed to evaluate stress and immune biomarkers in salivary samples of children with attention-deficit/hyperactivity disorder (ADHD) compared with those of healthy, non-ADHD children.

In our opinion, saliva may be a promising material for non-invasive ADHD monitoring. The results indicate that morning collection of saliva in contrast to unchanged salivary cortisol, may evaluate HPA and SNS dysregulations by measurements of sAA and immunoglobulins among ADHD children. 

## Figures and Tables

**Table 1 ijerph-18-00769-t001:** Inclusion and exclusion criteria for both groups.

Inclusion Criteria for the Study Group	Inclusion Criteria for the Control Group	Exclusion Criteria from the Study and Control Groups
Children of both sexes aged 6–11	Children of both sexes aged 6–11	Children with disorders of the central nervous system (e.g., epilepsy, serious injuries, and CNS infections)
Children with diagnosed ADHD in accordance with DSM-V diagnostic criteria (diagnosis confirmed separately by psychiatrist and clinical psychologist after a standardized and structured interview)	Lack of mental disorders—assessment with the use of a MINI-Kid questionnaire [25]	Co-existing: schizophrenia, bipolar affective disorder, or any serious somatic disorders
Clinically significant ADHD symptoms lasting over six months	A parent’s or legal guardian’s approval	Chronic somatic diseases
Children without hereditary mental disorders (first-degree relatives)		Persistent pharmacotherapy, hormonotherapy
A parent’s or legal guardian’s consent		Lack of acceptance from parents or legal guardians

ADHD—Attention Deficit Hyperactivity Disorder; DSM-V—*Diagnostic and Statistical Manual of Mental Disorders* (5th ed.), MINI-Kid–MINI International Neuropsychiatric Interview for Kids; CNS—Central Nervous System.

**Table 2 ijerph-18-00769-t002:** Anthropometric data and salivary concentrations of cortisol, alpha-amylase (sAA), and IgA, IgG, and IgM in the ADHD and control groups.

	Group	ADHD (n = 60) Mean ± SD Median (min–max)	Control (n = 72) Mean ± SD Median (min–max)	*p*-Value
Parameter				
Age (years)		8.6 ± 1.1 8 (6–11)	8.3 ± 1.2 8 (6–11)	ns
BMI z-score		0.6 ± 1.5	0.5 ± 1.2	ns
		0.4 (4.1 − (−2.6))	0.4 (3.5 − (−1.9))	
Waist size (cm)		63.9 ± 11 62 (49–101)	60.2 ± 6.6 60 (38–80)	ns
Hip size (cm)		76.1 ± 8.5 75 (57–104)	70.7 ± 6.3 70 (50–88)	0.001
Cortisol (ng/mL)		5.5 ± 4 4.3 (1.5–17.8)	5.5 ± 6.6 4.1 (0.2–54.2)	ns
sAA (U/mL)		112.6 ± 71.2 104.6 (11.7–296)	82.9 ± 49.8 76.1 (1.3–217.2)	0.030
IgA (ug/mL)		143.3 ± 72.7 160.8 (9.8–298.2)	115.5 ± 57.1 98.8 (22.3–235.4)	0.022
IgG (pg/mL)		4866.1 ± 1194.9 5352.5 (2032.3–6110)	5804 ± 2794.5 6013.1 (38.1–11,830.4)	0.017
IgM (ng/mL)		468.7 ± 115.3 469.6 (0.04–738.4)	414.4 ± 40.4 410.7 (318.6–556.8)	0.001

Statistical significance is given according to *p*-value with the statistical difference (*p* ≤ 0.05, *p* ≤ 0.01, *p* ≤ 0.001) vs. non-significant values in statistical analysis (ns). Abbreviations: n, number of patients; ADHD, patients diagnosed with attention-deficit/hyperactivity disorder; control group, healthy children. Statistical tests used were the Mann–Whitney U test. Abbreviations: Results are expressed as mean ± standard deviation, median (min–max ranges); n, number of examined children.

**Table 3 ijerph-18-00769-t003:** Spearman’s analysis of correlations (cortisol, alpha-amylase, IgA, IgG, IgM, BMI, and waist and hip sizes) between the ADHD and control groups.

**Control Group**								
	**IgA (ug/mL)**	**sAA** **(U/mL)**	**Cortisol** **(ng/mL)**	**IgG** **(pg/mL)**	**IgM** **(ng/mL)**	**BMI** **(kg/m^2^)**	**Waist Size (cm)**	**Hip Size (cm)**
IgA (ug/mL)		0.178	0.095	−0.117	−0.308	0.043	−0.047	0.065
sAA (U/mL)	0.178		0.161	−0.177	0.001	−0.017	−0.012	0.025
Cortisol (ng/mL)	0.095	0.161		−0.203	−0.047	−0.105	0.004	−0.046
IgG (pg/mL)	−0.117	−0.177	**−0.203 ***		0.090	0.077	−0.012	−0.060
IgM (ng/mL)	**−0.308 ***	0.001	−0.047	0.090		0.055	0.008	−0.019
BMI (kg/m^2^)	0.043	−0.017	−0.105	0.077	0.055		0.752	0.611
Waist size (cm)	−0.047	−0.012	0.004	−0.012	0.008	**0.752 ***		0.717
Hip size (cm)	0.065	0.025	−0.046	−0.060	−0.019	**0.611 ***	**0.717 ***	
**ADHD Group**								
	**IgA (ug/mL)**	**sAA** **(U/mL)**	**Cortisol** **(ng/mL)**	**IgG** **(pg/mL)**	**IgM** **(ng/mL)**	**BMI (kg/m^2^)**	**Waist Size (cm)**	**Hip Size (cm)**
IgA (ug/mL)		0.229	0.348	−0.467	0.175	−0.142	−0.049	−0.169
sAA (U/mL)	**0.229 ***		−0.119	−0.384	−0.118	−0.130	−0.189	−0.104
Cortisol (ng/mL)	**0.348 ***	−0.119		−0.031	−0.055	−0.350	−0.182	−0.282
IgG (pg/mL)	**−0.467 ***	**−0.384 ***	−0.031		−0.094	0.131	0.113	0.225
IgM (ng/mL)	0.175	−0.118	−0.055	−0.094		0.134	0.159	0.112
BMI (kg/m^2^)	−0.142	−0.130	**−0.350 ***	0.131	0.134		0.785	0.834
Waist size (cm)	−0.049	−0.189	−0.182	0.113	0.159	**0.785 ***		0.807
Hip size (cm)	−0.169	−0.104	**−0.282 ***	**0.225 ***	0.112	**0.834 ***	**0.807 ***	
**A force of correlation:**	**0.00–0.20**		**0.21–0.39**		**0.40–0.69**		**0.70–1.00**	

*p*-value in **bold** *, >0.05.

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
