# Peer review of "Stress/Immune Biomarkers in Saliva among Children with ADHD Status"

_ijerph, 2021, doi:10.3390/ijerph18020769_

Round 1

Reviewer 1 Report

I appreciate the opportunity to review this work. It presents a possible way to screen children with ADHD in a simple way and through the regular consultation of children with the dentist. The work is well designed and reaches conclusions that are well argued, however I think that it can be improved both in its content and in its presentation, so in Table 2 all the p values ​​should be put to three decimal places and when the p is <0.001 place only this value. In this way and how it is already considered in the section on material and methods, statistical studies, the significance is from p <0.05.
On the other hand, at these ages the BMI Z score for age and sex would be more appropriate than the BMI, and it is not necessary to take into account weight and height, since they are already included in this variable.
It would also be advisable, in order to be able to see the true importance of the variables that are significantly different, to perform a multivariate analysis, using binary logistic regression with respect to the main variable, healthy compared to patients with ADHD. This would give the true predictive value of the use of the indicated markers.

In principle, the work seems to me of interest, because as a strength it offers a possible fast, cheap way and taking advantage of the dental control visit, follow-up and control of children with hyperactivity, however, and for that reason I reported it with minor changes, that the data could be used much better with a more in-depth statistical analysis, specifically a multivariate analysis. It also indicated to them the need in pediatric ages to use, more than the BMI, the BMI Z score in relation to age and sex, given the large variations due to the times of growth in these ages.

Author Response

Dear Esteemed Reviewers 1,2,3 and 4

Thank you very much for your work and the reviewers’ comments. We have carefully addressed all comments provided by the reviewers. We feel that they have helped to improve our manuscript. In response to the comments, we made appropriate changes to the manuscript. All changes in the manuscript are highlighted in yellow. Please find our point-by-point response below.

Yours sincerely, on behalf of all authors,

Elzbieta Paszynska, DDS, MSc., PhD, Assoc. Prof.

(corresponding author)

Reviewer 1 Comments and Suggestions for Authors

I appreciate the opportunity to review this work. It presents a possible way to screen children with ADHD in a simple way and through the regular consultation of children with the dentist. The work is well designed and reaches conclusions that are well argued, however I think that it can be improved both in its content and in its presentation,

Answer: Thank you very much for the first comment, we also appreciate for the high estimation of the study value, especially among this group of children effected by ADHD.

 so in Table 2 all the p values ​​should be put to three decimal places and when the p is <0.001 place only this value. In this way and how it is already considered in the section on material and methods, statistical studies, the significance is from p <0.05.

Answer: We have presented p values to three decimal places and when the p is is <0.001 place only this value.

Answer: In regard to Reviewer 2 requiments: “…..In Table 2, the cortisol data should be marked as ns. In addition, an arterisk is not necessary, since all the other values are < 0.05.” I have removed asterisks and marked a cortisol p value as “ns”.

On the other hand, at these ages the BMI Z score for age and sex would be more appropriate than the BMI, and it is not necessary to take into account weight and height, since they are already included in this variable.

Answer: As suggested we have calculated and included   the BMI Z score to Table no. 2 and removed BMI, weight and height.

It would also be advisable, in order to be able to see the true importance of the variables that are significantly different, to perform a multivariate analysis, using binary logistic regression with respect to the main variable, healthy compared to patients with ADHD. This would give the true predictive value of the use of the indicated markers.

answer: We have a question about how to perform a multivariate analysisi. Does the reviewer mean the predictive value of these salivary  indicators in relation to z-score BMI or ADHD? We should consider a binary logistic regression.? zBMI is continuous, not 0-1, so we can only do multiple regression. Unless ADHD is the dependent variable. If you think that it is really needed we will perform in further analysisi.

In principle, the work seems to me of interest, because as a strength it offers a possible fast, cheap way and taking advantage of the dental control visit, follow-up and control of children with hyperactivity, however, and for that reason I reported it with minor changes, that the data could be used much better with a more in-depth statistical analysis, specifically a multivariate analysis. It also indicated to them the need in pediatric ages to use, more than the BMI, the BMI Z score in relation to age and sex, given the large variations due to the times of growth in these ages.

Answer:  We have calculated and included   the BMI z-score to Table no. 2. Unfortunately there were no statistical differences in contrast to BMI and body weight .

Reviewer 2 Report

The manuscript presented by Krahel et al., shows interesting results regarding the diagnosisi and knowledge of ADHD. Furthermore, the conclusions are clear, and correctly supported by the results. However, some metodological aspects, as well as some results, should be better explained for publication. Furthermore, it is necessary to correct some formal aspects. It would also be necessary to conclude with a proposal for a diagnostic method with the useful biomarkers.

It is not clear how is carried out the process of selection and dismissal of candidates.

In the text, the authors refer to figure 1, which does not appear in the manuscript.

There are fragments of the manuscript with letter with different type and size.

Regarding to numbers, sometimes appear writed by letters and sometimes as numbers. It must be standardized.

In table 3, control group, appears an asterisk in the table footer, but not in the table. In addition, in the ADHD group it is indicated in bold letters, not with an arterisks.

The BMI formula is not necessary, as it is well known. In any case, it is not squared (2 is down), and (m) should be in brackets, not height. 

In Table 2, the cortisol data should be marked as ns. In addition, an arterisk is not necessary, since all the other values are < 0.05.

Since all the parameters in section 2.4 are determined by ELISA, it could be indicated in the heading.

Some, but not all significant results, are listed in the first paragraph of section 3.1. Is there any reason?

It is not clear which parameters are considered significant in the correlation analysis (3.1.2). It is neccesary to explain the results obtained more clearly.

There are 70 references in the manuscript, and 71 in the list.

Author Response

The manuscript presented by Krahel et al., shows interesting results regarding the diagnosisi and knowledge of ADHD. Furthermore, the conclusions are clear, and correctly supported by the results. However, some metodological aspects, as well as some results, should be better explained for publication. Furthermore, it is necessary to correct some formal aspects. It would also be necessary to conclude with a proposal for a diagnostic method with the useful biomarkers.

It is not clear how is carried out the process of selection and dismissal of candidates.

Answer: All reasons of the dismissal of candidates - we have explained in Table 1 and Fig. 1 (Flow chart was prepared in numbers according to selection of subjects who fulfilled the criteria for inclusion), I have presented again below here  the Table 1 and Fig. 1  

Table 1. Inclusion and exclusion criteria for both groups.

Criteria for Inclusion into the Study Group

Criteria for Inclusion into the Control Group

Criteria for Exclusion from Study and Control Groups

Children of both sexes aged 6–11

Children of both sexes aged 6–11

Children with disorders of the central nervous system (e.g., epilepsy, serious injuries, and CNS infections)

Children with diagnosed ADHD in accordance with ICD-10 and DSM-V diagnostic criteria (diagnosis confirmed by two independent psychiatrists based on a standardized and structured interview)

Lack of mental disorders—assessment with the use of MINI-Kid questionnaire [71]

Co-existing:

schizophrenia,

bipolar affective disorder,

any serious somatic disorders

Clinically significant ADHD symptoms lasting over six months

A parent or legal guardian approval

Chronic somatic diseases

Children without hereditary mental disorders (first-degree relatives)

Persistent pharmacotherapy, hormonotherapy

A parent or legal guardian approval

Lack of acceptance from parents or legal guardians

ADHD - Attention Deficit Hyperactivity Disorder; ICD-10 - International Statistical Classification of Diseases and Related Health Problems (10th edition), DSM-V - Diagnostic and statistical manual of mental disorders (5th ed.), MINI-Kid - MINI International Neuropsychiatric Interview for Kids; CNS – Central Nervous System.

Figure 1. Flow chart  of  the study (please see also an pdf attachment)

Assessed for eligibility (n=520)

Excluded  (n=250)

¨   Not meeting inclusion criteria (n=250)

¨   Declined to participate (n=0)

¨   Other reasons (n= 0)

Lost to follow-up (alteration of ADHD to another diagnosis) (n=5)

Discontinued intervention (not enough cooperation by children) (n=2)

Allocated to intervention (n=120)

¨ Received allocated intervention (n=69)

¨ Did not receive allocated intervention (discontinued participation) (n=31)

Lost to follow-up (n=10)

Discontinued intervention (decline to participate) (n=18)

Allocated to intervention (n=150)

¨ Received allocated intervention (n=100)

¨ Did not receive allocated intervention (regular medication intake) (n=50)

Analysed  (n=72)
¨ Excluded from analysis  (n=0)

Allocation

Analysis

Follow-Up

Enrollment

Figure. 2

Analysed  (n=60)
¨ Excluded from analysis  (n=0)

 Salivary test (n=60)

¨   Declined to participate in test (n=2)

In the text, the authors refer to figure 1, which does not appear in the manuscript.

Answer: We would like to explain that Fig. 1 was previously included to Suppl material, because of trouble in appearing properly we have moved the Fig. 1 to the main material of the manuscript.

There are fragments of the manuscript with letter with different type and size.

Answer: We have checked again a type of letters (type and size) and it should be corrected in size 11. Anyway, in my opinion there is unexpected modification of letter size and type during the main manuscript transferring to MDPI system and it is behind of our manuscript preparation work.

Regarding to numbers, sometimes appear writed by letters and sometimes as numbers. It must be standardized.

Answer: Regarding to numbers we have rewritten to appear as numbers (they are highlighted in yellow color).

In table 3, control group, appears an asterisk in the table footer, but not in the table. In addition, in the ADHD group it is indicated in bold letters, not with an arterisks.

Answer: We have corrected again Table 3 and indicated as in bold p value with the asterisk

The BMI formula is not necessary, as it is well known. In any case, it is not squared (2 is down), and (m) should be in brackets, not height. 

Answer: The BMI formula was corrected (I apologize for the mistake) and (m) is in brackets, not height. We have replaced and added also BMI z-score results and calculation description is indicated in the same section of the Mat &Methods.

In Table 2, the cortisol data should be marked as ns. In addition, an arterisk is not necessary, since all the other values are < 0.05.

Answer: We have corrected to as suggested by the reviewer 1 and 2:

  • all asterisks are removed from tables
  • cortisol value is corrected as “ns” (in section Results I have added the p value with three decimals)

Since all the parameters in section 2.4 are determined by ELISA, it could be indicated in the heading.

Answer: I have indicated ELISA as a salivary samples measurements in the heading.

Some, but not all significant results, are listed in the first paragraph of section 3.1. Is there any reason?

Answer: The first paragraph of section 3.1 is rewritten again to clear the most significant information.

Revised text: lines 197-201:…….The mean age of ADHD children was 8.6±1.1 and controls 8.3±1.2 years with no statistically significant difference for age and height (p>0.05). ADHD participants weighted 10.5 % more than those in the control group (p≤0.01), as bodyweight  (p≤0.01), BMI (p≤0.009), hip circumference (p≤0.0002) were statistically different between patients and controls, but not waist size (p>0.05) (Table 2).

It is not clear which parameters are considered significant in the correlation analysis (3.1.2). It is neccesary to explain the results obtained more clearly.

Answer: We have modified  Table 3 and conclusion section,  

There are 70 references in the manuscript, and 71 in the list.

Answer: We have included totally 71 references, the last citation number 71 is indicated in Table 1 and referred to MINI KID questionnaire

Reviewer 3 Report

The article presents an important problem: Stress/immune biomarkers in saliva among children with ADHD status  

  • Abstract: please correct the significance levels, round them to three decimal places: eg no IgM (p = 0.0000001) only; p <0.001 and so on not p ≤ 0.0002) a p <0.001 and so on throughout the article.
  • Introduction: The etiology and occurrence are fully discussed Diagnosing and managing attention-deficit/hyperactivity disorder (ADHD). Well have been described a significant effect of stress on the activation of 59 hypothalamic-pituitary-adrenal (HPA) axis, resulting in increased cortisol production.
  • Material and Methods: this part is correct. There are errors in the description of somatic features, e.g. not height but it should be: body height (cm). Or it should be:       

                                    Body mass (kg)

                     BMI =    ………………

                                     Body height (m2)             

  • The study group and the control group were correctly selected. The statistical methods are appropriately selected and applied (The analyzed data were expressed as mean ± standard deviation, median, minimum and maximum values, as appropriate. Normality of distribution was tested using the Shapiro–Wilk test and equality of variances was checked using Levene's test. Comparison of two unpaired groups was performed using the Mann–Whitney U-test. The relationship between variables was analysed with Spearman's rank correlation coefficient. All results were considered significant at p<0.05. Statistical  analyses were performed with STATISTICA v13 software. 
  • Results: The research results are described here quite extensively. Tables 2,3 are correct and clearly present the obtained research results.
  • Discussion: this part of the work is correct. However, you can compare your own research even more to similar studies carried out.
  • Conclusions: this part of the article should be improved, conclusions should be related to the purpose of the work and answer research questions. Please refer to the purpose carefully: This cross-sectional study aimed to evaluate stress and immune 20 biomarkers in saliva samples of attention-deficit/hyperactivity disorder ADHD compared to 21 healthy non-ADHD children.
  • References: Reference can be completed.

The article is interesting, however, it requires some corrections. The manuscript, when revised, will deserve publication in the International Journal of Environmental Research and Public Health.

Author Response

Dear Esteemed Reviewers 1,2,3 and 4

Thank you very much for your work and the reviewers’ comments. We have carefully addressed all comments provided by the reviewers. We feel that they have helped to improve our manuscript. In response to the comments, we made appropriate changes to the manuscript. All changes in the manuscript are highlighted in yellow. Please find our point-by-point response below.

Yours sincerely, on behalf of all authors,

Elzbieta Paszynska, DDS, MSc., PhD, Assoc. Prof.

(corresponding author)

To Reviewer 3

The article presents an important problem: Stress/immune biomarkers in saliva among children with ADHD status  

Thank you very much for the first comment, we also appreciate for the high estimation of the study value, especially among this group of children effected by ADHD.

  • Abstract: please correct the significance levels, round them to three decimal places: eg no IgM (p = 0.0000001) only; p <0.001 and so on not p ≤ 0.0002) a p <0.001 and so on throughout the article.

Answer: In abstract and throughout the article (table 2 results) we have corrected the significance ad round them to three decimal places (when (p=0.0000001) only; (p<0.001).

  • Introduction: The etiology and occurrence are fully discussed Diagnosing and managing attention-deficit/hyperactivity disorder (ADHD). Well have been described a significant effect of stress on the activation of 59 hypothalamic-pituitary-adrenal (HPA) axis, resulting in increased cortisol production.

Answer: We are grateful for positive estimation of the reviewer once again. Our goal was to draw attention to the possibility of salivary markers utility in diagnosing and monitoring of children as a non-invasive method

  • Material and Methods: this part is correct. There are errors in the description of somatic features, e.g. not height but it should be: body height (cm). Or it should be:       

                                    Body mass (kg)

                     BMI =    ………………

                                     Body height (m2)             

Answer: The BMI formula was corrected (I apologize for the mistake) and (m) is in brackets, not height of course. According to Reviewers 1 and 2 suggestions we have replaced and added also BMI z-score results for both groups (please see Table 2). Unfortunately, in contrast to simple BMI and body weight there were no statistical differences according to added BMI z-score (p>0.05).

  • The study group and the control group were correctly selected. The statistical methods are appropriately selected and applied (The analyzed data were expressed as mean ± standard deviation, median, minimum and maximum values, as appropriate. Normality of distribution was tested using the Shapiro–Wilk test and equality of variances was checked using Levene's test. Comparison of two unpaired groups was performed using the Mann–Whitney U-test. The relationship between variables was analysed with Spearman's rank correlation coefficient. All results were considered significant at p<0.05. Statistical  analyses were performed with STATISTICA v13 software. 
  • Results: The research results are described here quite extensively. Tables 2,3 are correct and clearly present the obtained research results.
  • Discussion: this part of the work is correct. However, you can compare your own research even more to similar studies carried out.
  • Conclusions: this part of the article should be improved, conclusions should be related to the purpose of the work and answer research questions. Please refer to the purpose carefully: This cross-sectional study aimed to evaluate stress and immune 20 biomarkers in saliva samples of attention-deficit/hyperactivity disorder ADHD compared to 21 healthy non-ADHD children.

Answer: This study's principal objective was then to compare the level of several biomarkers of stress/immune responses (cortisol, sAA, sIgA, IgG, IgM) among children with a clinically confirmed ADHD vs. non-affected ADHD and healthy children. Our results suggest that morning collection of saliva in contrast to cortisol would be useful for sAA as stress biomarker and immunoglobulins measurements among ADHD children.

  • References: Reference can be completed.

Answer: Originallyy we have included totally 71 references, the last citation number 71 is indicated in Table 1 and referred to MINI KID questionnaire

The article is interesting, however, it requires some corrections. The manuscript, when revised, will deserve publication in the Intern. J Enr.Res Pub Health.

Answer: We are really grateful for such posotove feedback of our clinical and biochemical project based on salivary samples. We believe that our work will be useful in future projects for other researchers.

Reviewer 4 Report

In this manuscript, entitled “Stress/immune biomarkers in saliva among children with ADHD status,” the authors attempted to examine stress and immune biomarkers in saliva samples of attention-deficit/hyperactivity disorder ADHD.  

The authors performed clinical evaluation including physical parameters, saliva collection and measurements of cortisol, alpha-amylase sAA, and secreted immunoglobulins such as sIgA, IgG, and IgM by ELISA. In general, the manuscript is well-written; however, this manuscript has a minor critic that should be corrected.

  1. line 133> The authors used formula to calculate BMI “BMI =weight (kg)/(height)2m”. In this formula “2” should be superscripted.

Author Response

Dear Esteemed Reviewers 1,2,3 and 4

Thank you very much for your work and the reviewers’ comments. We have carefully addressed all comments provided by the reviewers. We feel that they have helped to improve our manuscript. In response to the comments, we made appropriate changes to the manuscript. All changes in the manuscript are highlighted in yellow. Please find our point-by-point response below.

Yours sincerely, on behalf of all authors,

Elzbieta Paszynska, DDS, MSc., PhD, Assoc. Prof.

(corresponding author)

Answer To Reveiwer 4:

Comments and Suggestions for Authors

In this manuscript, entitled “Stress/immune biomarkers in saliva among children with ADHD status,” the authors attempted to examine stress and immune biomarkers in saliva samples of attention-deficit/hyperactivity disorder ADHD.  

The authors performed clinical evaluation including physical parameters, saliva collection and measurements of cortisol, alpha-amylase sAA, and secreted immunoglobulins such as sIgA, IgG, and IgM by ELISA. In general, the manuscript is well-written; however, this manuscript has a minor critic that should be corrected.

  1. line 133> The authors used formula to calculate BMI “BMI =weight (kg)/(height)2m”. In this formula “2” should be superscripted.

Answer: The BMI formula was corrected (I apologize for the mistake) and (m) is in brackets, not height of course. As sugested the Reviewers 1 and 2 we have replaced and added also BMI z-score results and calculation description is indicated in the same section of the Mat &Methods. Unfortunately a new index is not statistical different (p>0.05) as the previous BMI and body weight, please see Table 2.